# Public preferences towards data management and governance in Swiss biobanks: results from a nationwide survey

Caroline Brall [1], Claudia Berlin [2], Marcel Zwahlen [2], Effy Vayena [1], Matthias Egger [2,3], Kelly E Ormond [1]

¹Department of Health Sciences and Technology, ETH Zurich, Zurich, Switzerland
²Institute of Social and Preventive Medicine, University of Bern, Bern, Switzerland
³Population Health Sciences, Bristol Medical School, University of Bristol, Bristol, UK

**Correspondence to**
Dr Caroline Brall;
carobrall@gmail.com

## ABSTRACT

**Objectives** This article aims to measure the willingness of the Swiss public to participate in personalised health research, and their preferences regarding data management and governance.

**Setting** Results are presented from a nationwide survey of members of the Swiss public.

**Participants** 15106 randomly selected Swiss residents received the survey in September 2019. The response rate was 34.1% (n=5156). Respondent age ranged from 18 to 79 years, with fairly uniform spread across sex and age categories between 25 and 64 years.

**Primary and secondary outcome measures** Willingness to participate in personalised health research and opinions regarding data management and governance.

**Results** Most respondents preferred to be contacted and reconsented for each new project using their data (39%, 95% CI: 37.4% to 40.7%), or stated that their preference depends on the project type (29.4%, 95% CI: 27.9% to 31%). Additionally, a majority (52%, 95% CI: 50.3% to 53.8%) preferred their data or samples be stored anonymously or in coded form (43.4%, 95% CI: 41.7% to 45.1%). Of those who preferred that their data be anonymised, most also indicated a wish to be recontacted for each new project (36.8%, 95% CI: 34.5% to 39.2%); however, these preferences are in conflict. Most respondents desired to personally own their data. Finally, most Swiss respondents trust their doctors, along with researchers at universities, to protect their data.

**Conclusion** Insight into public preference can enable Swiss biobanks and research institutions to create management and governance strategies that match the expectations and preferences of potential participants. Models allowing participants to choose how to interact with the process, while more complex, may increase individual willingness to provide data to biobanks.

## INTRODUCTION

Personalised health strives to improve clinical outcomes by providing more effective prevention and therapy for individuals, thus improving morbidity and mortality. Large amounts of health data and samples are necessary to achieve the goals of personalised medicine, requiring that individuals participate in

### STRENGTHS AND LIMITATIONS OF THIS STUDY

⇒ This nationwide study examined preferences of 5156 members of the Swiss public regarding data management and governance when hypothetically providing data or samples for personalised health research.

⇒ We were able to construct non-response weights by all sociodemographic characteristics, making the results reasonably generalisable to the Swiss population.

⇒ Respondents were asked about type of consent and storage preferences in two separate questions, making it impossible to determine which one is more important to respondents.

biomedical research by donating their health data and samples, and that the samples and data be available for reuse in future research studies.[1] Studies have shown that public preferences around sharing personal health and genetic information in a research setting vary depending on societal, demographic and jurisdictional context.[2] In Switzerland, we previously found that the public generally holds a positive opinion towards personalised health research, and that a majority are hypothetically willing to donate data and samples for such projects.[3] Yet, in biobank research, concerns about data privacy and reuse are closely linked to public willingness to participate in research.[3–6]

Two research design components greatly influence the ability to reuse samples and data for future research, and participants' willingness to provide samples.[7] First, a biobank can choose to store data in an identifiable, coded or anonymous form. If data and samples are stored and labelled with identifiable information, they can easily be linked back to donors. In contrast, anonymous data storage removes all personal identifiers from the data; linking back to a specific person is possible, but

only with tremendous effort. An intermediate option is storage in coded form, which links an individual's data and samples via a securely stored code. Second, the way in which informed consent occurs can influence participant trust and the feasibility of sample and data reuse.[8 9] Broad consent (also known as general consent) denotes that individuals consent to their data and samples being used in future unspecified research studies. Broad consent approaches increase data availability, thereby facilitating future research, but rely on potential participants' comfort with agreeing to take part in future research without knowing what areas it will address.[10] On the other hand, specific consent requires that individuals provide consent for every study that explicitly uses their data or samples. Other more specific consent models exist beyond those presented, including blanket consent, presumed consent with opt-out or tiered consent (for a detailed description, see [11–13]). While study-specific consent is often technically challenging and resource-intensive to implement and update, it allows participants greater control over their samples and data, particularly in the case of potentially stigmatising or controversial research. Finally, choices in the research design for data storage and consent approach influence how data can be used, whether individuals must or can be recontacted for consent, and possibilities for reporting individual results.[14]

In Switzerland, existing biobanks are primarily linked to five university hospitals and generally use a broad consent approach.[15 16] Personalised health initiatives are scaling up, and broad consent has been promoted at the national level.[16] However, we do not know how the Swiss public views this type of consent, and what other views and expectations they may hold concerning the management and governance of their health data and samples. A better understanding of what potential participants expect from personalised health biobanks and related research infrastructure, and the data management and governance practices that influence willingness to donate data and samples, will allow biobanks to address the needs of the public in the early stages of project development.

This paper presents findings from a nationwide survey[3] of the Swiss public's preferences for data management and governance when hypothetically providing data or samples for personalised health research. We report preferences for consent type, data storage, ownership and management responsibilities, and information respondents need before donating data or samples. Finally, we present respondents' levels of trust in various actors across the health data ecosystem.

## METHODS

We conducted a cross-sectional survey, with full details of the survey development and pilot testing published previously.[3] Briefly, the survey contained questions to assess individual attitudes, concerns and expectations towards hypothetically providing health data or biological samples for personalised health research. It consisted of six parts: general attitudes; motivations and barriers to participating in personalised health research; expectations towards data management; data governance; data sharing and uses; and willingness to receive results. The questionnaire was available in English, German, Italian and French, and consisted of 23 closed (binary, 5-point Likert scale and multiple choice) questions. This paper will focus on the findings for data management, data governance and data sharing.

The survey was mailed to 15 106 Swiss residents over the age of 18 years in September 2019. According to the Swiss Statistical Survey Ordinance, the Swiss Federal Statistical Office (FSO) provided the stratified random sample, which covered the three main language regions (SR 431.012.1, article 13c). The sampled potential participants received the survey by regular mail in their language of correspondence (indicated at their municipality). The survey could be completed either on paper or through a web link on the Qualtrics survey platform (Qualtrics, Provo, Utah, USA); both options allowed mapping of individual responses with sociodemographic characteristics provided by the FSO. Participants received two reminders (after 3 and 7 weeks). Responses were collected over 20 weeks from September 2019 to January 2020. By responding to the survey, participants provided their informed consent.

### Patient and public involvement

The development of the research question and outcome measures was informed by the aim to learn more about public preferences and expectations towards donating health data or samples for personalised health research. Seventeen members of the public were involved in testing preliminary versions of the questionnaire, which was adapted according to their comments.

As we indicated to respondents that they would not be contacted again after completing the survey, results were not actively disseminated to participants, but made available upon request. In addition, the results are published open access.

### Statistical analysis

After evaluating the data for completeness (minimum of 50% data completeness for inclusion in analysis), we analysed it using STATA (V.15, College Station, Texas, USA). We linked the survey data with demographic data from the FSO (gender, age, language, household size, nationality, marital status and municipality of residence) using the FSO unique identifiers. To account for differences between survey respondents and the general population of Switzerland, we applied survey weights using gender, age and language region, and included additional variables provided by the FSO. Our final dataset included age, gender, nationality, number of household members, marital status, having biological children, language region, type of municipality of residence (urban or rural), education, religiosity, current or previous employment in the health sector, health status and type of survey

response (online or paper based). We excluded missing data and used relative proportions for statistical analysis, as they allow a more straightforward interpretation than ORs: comparing a proportion of 60% with one of 30% leads to a relative proportion of 2.0, but an OR of 3.5. We used modified multivariable Poisson regression to analyse the relative proportion of respondents' willingness to provide health data or biological samples for personalised health research, and to adjust for the other respondent characteristics.[17] To simplify interpretation of the results, we collapsed a 5-point Likert scale (1=no trust to 5=strong trust) to binary variables (1–3=low trust, 4–5=strong trust).

## RESULTS
### Sample description
As reported in detail elsewhere,[3] 5086 complete responses were received, representing an overall response rate of 34.1%. Most responses (71.0%) were submitted via the web-based survey platform. Respondent age ranged from 18 to 79 years, with fairly even spread across sex and age categories between 25 and 64 years (table 1). A majority of respondents were Swiss nationals (76.0%), lived in a German-speaking region (70.8%), in urban areas (61.0%), were married (50.9%), had children (57.6%) and lived in households with three to five persons (43.7%). Most had secondary education (65.2%) and described their health status as somewhat (47.1%) or very healthy (36.6%); a majority had never worked in the health sector (79.8%). Thirteen per cent reported being very religious. More than half of respondents (53.6%) expressed a hypothetical willingness to provide health data or biological samples for personalised health research purposes.[3]

### Preferences on types of consent
Regarding consent to use data from a hypothetical biobank, 39.0% (95% CI: 37.4% to 40.7%) of respondents wished to be asked for consent for each new project (study-specific consent). Furthermore, 29.4% (95% CI: 27.9% to 31.0%) indicated that this would depend on the type of project, and 17.7% (95% CI: 16.4% to 19.0%) preferred to be asked only once when donating data or samples (general consent). A minority of respondents (13.8%, 95% CI: 12.7% to 15.1%) were unsure of their preference. Compared with those who wished to be asked only once, participants who wished to be asked for permission for each project or who were unsure were less willing to hypothetically provide data and samples for research (adjusted relative proportion (aRP)=0.86, 95% CI: 0.79 to 0.93; aRP=0.35, 95% CI: 0.29 to 0.43) (figure 1). Table 1 shows demographic information for different types of consent. The proportion of those who were more inclined to be asked only once (general consent) varied for some naturally ordered characteristics but not for all when testing for a trend: age groups (p for trend<0.001), number of household members (p for trend=0.124), religiousness

(p for trend=0.770), education (p for trend=0.003) and health status (p for trend=0.598).

### Preferences for data storage
When asked how they preferred their data to be stored, most respondents preferred their data or samples to be anonymised (52.0%, 95% CI: 50.3% to 53.8%) or stored and used in coded form (43.4%, 95% CI: 41.7% to 45.1%). Only a few respondents (4.5%, 95% CI: 3.9% to 5.3%) preferred their data be stored and used in an identifiable form (figure 2). Respondents who preferred coded or identifiable data and sample storage were more willing to hypothetically donate data or samples (aRP: 1.57, 95% CI: 1.47 to 1.68; aRP: 1.67, 95% CI: 1.47 to 1.89) than those who preferred anonymised samples. Many respondents who would like their data to be stored anonymously indicated a preference to be recontacted for each new project (36.8%, 95% CI: 34.5% to 39.2%) or recontacted depending on the new project type (26.5%, 95% CI: 24.4% to 28.6%).

### Information participants need before deciding to donate data and samples to a biobank
We asked all respondents to endorse the top three pieces of information they would need to decide whether to donate data and samples to a biobank (figure 3). While nearly all options had the endorsement of at least 25.0% of participants, the most relevant pieces of information were 'the exact types of research which will be conducted' (57.1%) and 'who has access to my data and samples' (48.3%). Only a few participants indicated that none of this information would help them decide (7.4%). Respondents were more willing to hypothetically provide data or samples when they endorsed any of the following pieces of information: 'the exact types of research which will be conducted', 'the potential benefits and risks of donating my data and samples', 'the way data and samples are stored, such as made anonymous, encoded or stored together with my name', and 'the security measures to keep data and samples private and protected'. The answer options 'who will benefit from the research' and 'none of this information would help me decide' were negatively associated with willingness to donate data or samples to a biobank.

### Importance of financial or other material compensation for data donation
Most respondents indicated that money or other material compensation would not/would rather not be important (56.9%) for deciding whether to participate in a publicly funded Swiss biobank. Remaining participants were split, rating compensation as either moderately (24.9%) or rather/very important (18.3%). The greater importance a respondent placed on financial compensation, the less willing this person was to report hypothetical willingness to participate in a biobank (aRP: rather not important: 1.24, 95% CI: 1.14 to 1.34; very important: 0.79, 95% CI: 0.67 to 0.92).

**Table 1** Preferences for consent type according to sociodemographic factors (all weighted proportions except N)

| | Sample | Population | How often should a biobank ask for permission? | | | |
| | | | Ask me only once, when I donate my data or samples | Ask me again for every new project | It would depend on the type of project that is being considered | Not sure |
| | N | % | % (95% CI) | % (95% CI) | % (95% CI) | % (95% CI) |
|---|---|---|---|---|---|---|
| **Total** | **5086** | **100** | **17.7 (16.4 to 19.0)** | **39.0 (37.4 to 40.7)** | **29.4 (27.9 to 31.0)** | **13.8 (12.7 to 15.1)** |
| **Age group** | | | | | | |
| 18–24 | 594 | 8.6 | 10.8 (8.2 to 14.1) | 41.8 (37.0 to 46.7) | 34.7 (30.1 to 39.6) | 12.7 (9.8 to 16.4) |
| 25–34 | 758 | 17.7 | 14.5 (11.7 to 17.9) | 41.8 (37.6 to 46.2) | 30.2 (26.3 to 34.4) | 13.4 (10.7 to 16.7) |
| 35–44 | 632 | 18.5 | 15.4 (12.3 to 19.1) | 46.8 (42.2 to 51.5) | 26.3 (22.4 to 30.6) | 11.5 (8.9 to 14.8) |
| 45–54 | 927 | 19.5 | 18.2 (15.4 to 21.4) | 41.3 (37.6 to 45.1) | 27.5 (24.2 to 31.1) | 13.0 (10.6 to 15.9) |
| 55–64 | 1005 | 17.5 | 17.4 (14.8 to 20.4) | 37.6 (34.0 to 41.4) | 29.6 (26.3 to 33.2) | 15.3 (12.8 to 18.3) |
| 65–74 | 857 | 13.1 | 26.8 (23.3 to 30.5) | 27.4 (24.0 to 31.1) | 29.7 (26.2 to 33.4) | 16.1 (13.5 to 19.2) |
| 75–79 | 313 | 5.2 | 25.1 (19.7 to 31.5) | 21.3 (16.3 to 27.2) | 35.5 (29.3 to 42.2) | 18.1 (13.5 to 23.8) |
| Total | 5086 | 100 | 17.7 (16.4 to 19.0) | 39.0 (37.4 to 40.7) | 29.4 (27.9 to 31.0) | 13.8 (12.7 to 15.1) |
| **Sex** | | | | | | |
| Male | 2451 | 50.1 | 18.8 (17.0 to 20.8) | 38.3 (35.9 to 40.8) | 29.5 (27.3 to 31.8) | 13.4 (11.8 to 15.2) |
| Female | 2635 | 49.9 | 16.6 (14.9 to 18.4) | 39.8 (37.5 to 42.1) | 29.4 (27.3 to 31.6) | 14.3 (12.7 to 16.0) |
| Total | 5086 | 100 | 17.7 (16.4 to 19.0) | 39.0 (37.4 to 40.7) | 29.4 (27.9 to 31.0) | 13.8 (12.7 to 15.1) |
| **Nationality** | | | | | | |
| Swiss | 4216 | 76.0 | 18.6 (17.2 to 20.0) | 39.3 (37.5 to 41.1) | 29.9 (28.2 to 31.6) | 12.3 (11.2 to 13.5) |
| Non-Swiss | 870 | 24.0 | 15.0 (12.3 to 18.1) | 38.2 (34.3 to 42.3) | 28.1 (24.5 to 32.0) | 18.7 (15.7 to 22.0) |
| Total | 5086 | 100 | 17.7 (16.4 to 19.0) | 39.0 (37.4 to 40.7) | 29.4 (27.9 to 31.0) | 13.8 (12.7 to 15.1) |
| **Number of household members** | | | | | | |
| 1 | 760 | 18.1 | 18.1 (14.9 to 21.9) | 36.4 (32.2 to 40.9) | 31.4 (27.4 to 35.8) | 14.0 (11.3 to 17.2) |
| 2 | 1854 | 35.5 | 19.5 (17.4 to 21.7) | 37.8 (35.1 to 40.6) | 29.9 (27.4 to 32.5) | 12.7 (11.0 to 14.7) |
| 3–5 | 2347 | 43.7 | 16.2 (14.4 to 18.1) | 41.2 (38.7 to 43.7) | 28.2 (26.0 to 30.5) | 14.4 (12.7 to 16.3) |
| 6 persons and more | 125 | 2.8 | 15.9 (9.3 to 25.7) | 36.5 (27.2 to 47.0) | 29.5 (20.7 to 40.1) | 18.1 (11.1 to 28.1) |
| Total | 5086 | 100 | 17.7 (16.4 to 19.0) | 39.0 (37.4 to 40.7) | 29.4 (27.9 to 31.0) | 13.8 (12.7 to 15.1) |
| **Marital status** | | | | | | |
| Single | 1669 | 35.3 | 13.0 (11.1 to 15.2) | 44.8 (41.8 to 47.9) | 30.0 (27.3 to 32.9) | 12.2 (10.3 to 14.3) |
| Married | 2733 | 50.9 | 19.8 (18.0 to 21.7) | 37.2 (35.0 to 39.5) | 28.1 (26.1 to 30.2) | 14.8 (13.3 to 16.6) |
| Widowed | 150 | 3.1 | 22.7 (15.3 to 32.4) | 22.2 (15.3 to 31.0) | 34.0 (25.6 to 43.6) | 21.0 (14.6 to 29.4) |
| Divorced | 534 | 10.7 | 21.9 (17.9 to 26.6) | 33.1 (28.4 to 38.1) | 32.3 (27.8 to 37.2) | 12.7 (9.7 to 16.4) |
| Total | 5086 | 100 | 17.7 (16.4 to 19.0) | 39.0 (37.4 to 40.7) | 29.4 (27.9 to 31.0) | 13.8 (12.7 to 15.1) |
| **Biological children** | | | | | | |
| Yes | 2968 | 57.6 | 19.6 (17.9 to 21.4) | 36.9 (34.8 to 39.1) | 28.8 (26.8 to 30.8) | 14.7 (13.3 to 16.4) |
| No | 2074 | 42.4 | 15.1 (13.3 to 17.1) | 41.8 (39.1 to 44.5) | 30.5 (28.0 to 33.1) | 12.6 (11.0 to 14.5) |
| Missing | 44 | | | | | |
| Total | 5086 | 100 | 17.7 (16.4 to 19.0) | 39.0 (37.3 to 40.7) | 29.5 (28.0 to 31.1) | 13.8 (12.7 to 15.1) |
| **Language region** | | | | | | |
| German | 2257 | 70.8 | 17.9 (16.3 to 19.6) | 40.0 (37.9 to 42.2) | 29.9 (28.0 to 32.0) | 12.2 (10.8 to 13.7) |
| French | 1366 | 24.5 | 17.3 (15.3 to 19.5) | 35.7 (33.0 to 38.4) | 28.3 (25.7 to 30.9) | 18.8 (16.6 to 21.1) |
| Italian | 1463 | 4.7 | 16.9 (15.0 to 19.1) | 41.7 (39.0 to 44.3) | 27.9 (25.6 to 30.4) | 13.5 (11.7 to 15.5) |
| Total | 5086 | 100 | 17.7 (16.4 to 19.0) | 39.0 (37.4 to 40.7) | 29.4 (27.9 to 31.0) | 13.8 (12.7 to 15.1) |
| **Urban/rural municipality** | | | | | | |

**Table 1** Continued

| | Sample | Population | Ask me only once, when I donate my data or samples | Ask me again for every new project | It would depend on the type of project that is being considered | Not sure |
|---|---|---|---|---|---|---|
| | N | % | % (95% CI) | % (95% CI) | % (95% CI) | % (95% CI) |
| Urban | 3104 | 61.0 | 17.5 (15.9 to 19.3) | 39.5 (37.3 to 41.7) | 29.5 (27.5 to 31.6) | 13.4 (12.0 to 15.0) |
| Intermediary | 1086 | 21.6 | 17.5 (15.0 to 20.3) | 38.8 (35.3 to 42.4) | 28.7 (25.6 to 32.0) | 15.0 (12.6 to 17.8) |
| Rural | 896 | 17.3 | 18.6 (15.7 to 21.9) | 37.6 (33.8 to 41.6) | 30.0 (26.4 to 33.8) | 13.8 (11.3 to 16.8) |
| Total | 5086 | 100 | 17.7 (16.4 to 19.0) | 39.0 (37.4 to 40.7) | 29.4 (27.9 to 31.0) | 13.8 (12.7 to 15.1) |
| **Education** | | | | | | |
| Compulsory education or less | 385 | 8.4 | 12.1 (8.7 to 16.6) | 28.0 (22.8 to 33.8) | 24.5 (19.6 to 30.2) | 35.4 (29.7 to 41.5) |
| Upper secondary education | 3394 | 65.2 | 17.4 (15.9 to 19.0) | 37.6 (35.5 to 39.6) | 30.8 (28.9 to 32.8) | 14.3 (12.9 to 15.8) |
| Tertiary education | 1283 | 26.4 | 20.1 (17.5 to 23.0) | 46.2 (42.8 to 49.7) | 27.8 (24.8 to 30.9) | 5.9 (4.6 to 7.6) |
| Missing | 24 | | | | | |
| Total | 5086 | 100 | 17.7 (16.4 to 19.0) | 39.1 (37.4 to 40.8) | 29.5 (27.9 to 31.1) | 13.8 (12.7 to 15.0) |
| **Religiousness** | | | | | | |
| Very much | 695 | 13.0 | 16.5 (13.3 to 20.2) | 38.1 (33.6 to 42.8) | 27.2 (23.3 to 31.6) | 18.2 (14.9 to 22.0) |
| Somewhat | 2217 | 43.6 | 18.2 (16.3 to 20.2) | 35.4 (33.0 to 38.0) | 31.2 (28.8 to 33.6) | 15.2 (13.4 to 17.2) |
| Not at all | 2138 | 43.5 | 17.7 (15.7 to 19.8) | 42.8 (40.2 to 45.4) | 28.5 (26.2 to 31.0) | 11.0 (9.5 to 12.7) |
| Missing | 36 | | | | | |
| Total | 5086 | 100 | 17.7 (16.5 to 19.1) | 39.0 (37.3 to 40.7) | 29.5 (28.0 to 31.1) | 13.8 (12.6 to 15.0) |
| **Working in health sector?** | | | | | | |
| Yes | 1010 | 20.2 | 17.7 (15.0 to 20.8) | 44.7 (40.9 to 48.6) | 28.9 (25.5 to 32.5) | 8.6 (6.7 to 11.0) |
| No | 4063 | 79.8 | 17.7 (16.3 to 19.2) | 37.6 (35.8 to 39.5) | 29.6 (27.9 to 31.4) | 15.0 (13.7 to 16.4) |
| Missing | 13 | | | | | |
| Total | 5086 | 100 | 17.7 (16.5 to 19.1) | 39.1 (37.4 to 40.8) | 29.5 (27.9 to 31.1) | 13.7 (12.6 to 14.9) |
| **Health status** | | | | | | |
| Very unhealthy | 61 | 0.8 | 20.9 (10.2 to 37.9) | 28.1 (15.5 to 45.4) | 26.0 (12.5 to 46.3) | 25.1 (12.5 to 44.0) |
| Somewhat unhealthy | 115 | 2.0 | 13.5 (7.9 to 22.2) | 29.8 (20.6 to 41.1) | 26.4 (17.5 to 37.7) | 30.3 (21.2 to 41.2) |
| Neutral | 661 | 13.5 | 17.5 (14.4 to 21.2) | 33.0 (28.7 to 37.6) | 31.0 (26.8 to 35.5) | 18.5 (15.1 to 22.5) |
| Somewhat healthy | 2578 | 47.1 | 18.7 (16.8 to 20.7) | 36.6 (34.3 to 39.0) | 31.1 (28.9 to 33.4) | 13.6 (12.1 to 15.4) |
| Very healthy | 1643 | 36.6 | 16.7 (14.6 to 18.9) | 45.1 (42.2 to 48.0) | 27.1 (24.6 to 29.8) | 11.1 (9.4 to 13.0) |
| Missing | 28 | | | | | |
| Total | 5086 | 100 | 17.7 (16.4 to 19.0) | 39.0 (37.4 to 40.7) | 29.5 (27.9 to 31.1) | 13.8 (12.6 to 15.0) |
| **Type of response** | | | | | | |
| Web based | 3519 | 71.0 | 17.6 (16.1 to 19.2) | 41.7 (39.7 to 43.7) | 29.0 (27.2 to 30.9) | 11.7 (10.4 to 13.1) |
| Paper based | 1567 | 29.0 | 17.9 (15.6 to 20.5) | 32.0 (29.1 to 35.1) | 30.6 (27.7 to 33.6) | 19.5 (17.1 to 22.0) |
| Total | 5086 | 100 | 17.7 (16.4 to 19.0) | 39.0 (37.4 to 40.7) | 29.4 (27.9 to 31.0) | 13.8 (12.7 to 15.1) |

## Preferences for data ownership

Nearly half of respondents indicated that they should personally own the data and samples donated to a biobank (49.5%, 95% CI: 47.8% to 51.2%). Other potential 'data owners' included universities involved with the biobank (11.0%, 95% CI: 10.0% to 12.1%), the biobank (10.7%, 95% CI: 9.7% to 11.8%), specific researchers who make discoveries (9.5%, 95% CI: 8.5% to 10.5%) or the Swiss government (5.4%, 95% CI: 4.7% to 6.3%). Participants who felt that ownership should belong to the biobank (aRP: 1.32, 95% CI: 1.21 to 1.45), the Swiss government (aRP: 1.23, 95% CI: 1.08 to 1.39), universities involved with the biobank (aRP: 1.36, 95% CI: 1.24 to 1.48) and specific researchers who make discoveries (aRP: 1.31, 95% CI: 1.18 to 1.45) were significantly more likely to be willing to hypothetically participate in a biobank,

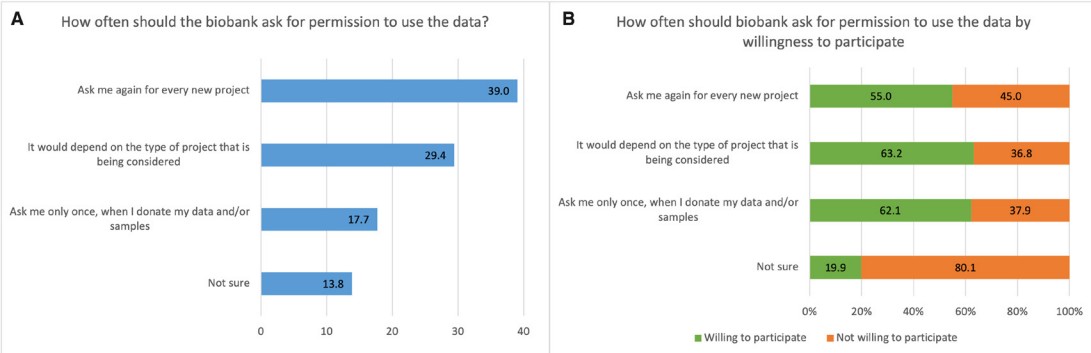

**Figure 1** Overview of how often the biobank should ask for permission to reuse the data (type of consent). (A) Overall percentages of respondents who preferred various options regarding how often the biobank should ask for permission to reuse the data (type of consent). 'Ask me only once' corresponds to a 'broad consent' approach, and 'Ask me again for every new project' corresponds to specific consent and/or a dynamic consent approach. (B) Per cent of each response choice who is willing/not willing to participate in personalised health research.

compared with persons who preferred that data ownership belong to them personally. Those that endorsed ownership by 'no one' (6.2%, 95% CI: 5.4% to 7.1%) were less likely to express willingness to participate (aRP: 0.76, 95% CI: 0.63 to 0.92).

### Views on responsibility for data management and storage

Concerning data management and storage, most respondents preferred governance by the biobank (ie, the management board, 34.3%, 95% CI: 32.7% to 35.9%) or an independent expert committee (eg, independent researchers: scientists and clinicians not associated with the biobank, 30.7%, 95% CI: 29.1% to 32.3%). Those who favoured an independent committee representing the public (eg, citizens, patients, the public) (8.0%) were less willing to participate in a hypothetical biobank (aRP: 0.78, 95% CI: 0.67 to 0.91) compared with respondents preferring governance by the biobank.

### Trusted actors

Figure 4 shows the extent to which respondents trusted different actors to keep their data and samples confidential and protected if they had access to them in a biobank. Most respondents reported the strongest trust in their own doctor (76.3%), medical doctors in general (41.9%) and researchers at a university (41.7%). Health insurers (8.4%), pharmaceutical companies (6.4%), other

for-profit companies from Switzerland (3.4%) and other global for-profit companies (2.0%) are trusted least.

Reported trust in researchers differed by level of education. Those with tertiary education reported stronger trust in researchers at a university (aRP: 1.26, 95% CI: 1.15 to 1.39) or researchers at other public institutes (aRP: 1.53, 95% CI: 1.34 to 1.75) than did those with upper secondary education. Trust in health insurers differed by language region, education, religion and health status: respondents living in the Italian-speaking region (aRP: 0.56, 95% CI: 0.42 to 0.75) of Switzerland trusted health insurance companies significantly less than respondents from the German-speaking region. Trust in health insurers decreased as educational level increased (compulsory education or less: aRP: 1.51, 95% CI: 1.07 to 2.14; tertiary education: aRP: 0.67, 95% CI: 0.47 to 0.94), but increased as self-reported religiosity decreased (somewhat religious: aRP: 0.68, 95% CI: 0.51 to 0.92; not at all religious: aRP: 0.45, 95% CI: 0.32 to 0.63). Finally, healthier respondents' trust in health insurers was lower than for persons reporting a very unhealthy health status (somewhat healthy: aRP: 0.33, 95% CI: 0.14 to 0.75; very healthy: aRP: 0.40, 95% CI: 0.17 to 0.92).

Strong trust in eight of the nine mentioned institutions (see figure 4) increased willingness to participate in a hypothetical personalised health research study or

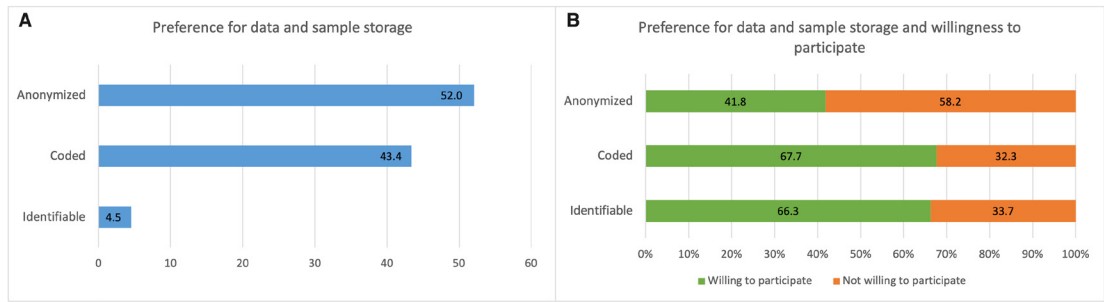

**Figure 2** Preferences for data and sample storage. (A) Percentages of respondents who preferred each approach to data and sample storage; (B) per cent of each response choice who is willing/not willing to participate in personalised health research.

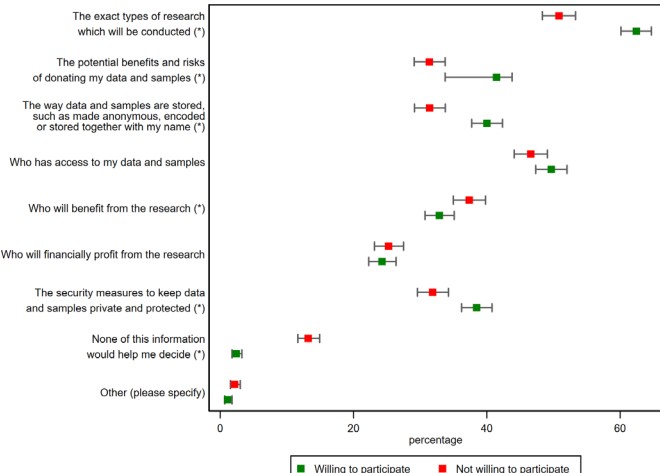

**Figure 3** Preferences for information necessary in order to make a decision to donate data and samples to a biobank. Figure presents population-weighted percentages; respondents were asked to indicate their top three informational preferences. * indicates statistical difference between the two groups of respondents willing and not willing to participate, p<0.05.

biobank. Only trust in global private, for-profit companies did not influence willingness to participate in such research. Respondents also distinguished between whether for-profit companies are from Switzerland or not: respondents with strong trust in Swiss for-profit companies were more willing to participate in personalised health research (aRP: 1.33, 95% CI: 1.18 to 1.51) compared with respondents with low trust in these companies. Those with compulsory education or less were more likely to trust Swiss for-profit companies (aRP: 2.29, 95% CI: 1.35 to 3.85) than those with secondary or tertiary education.

## DISCUSSION

This survey assessed the preferences of the Swiss public regarding management and governance of health data and biological samples in biobanks for personalised health research. We found that a majority of respondents hypothetically prefer anonymous or coded data storage in a biobank (52% and 43%, respectively), yet also prefer to be recontacted for each new project that will use their data (39%) or state that their preference depends on the type of project being considered (29%). Material compensation is not important in exchange for providing data and samples to a Swiss publicly funded biobank, but nearly 50% prefer to own their own data. Respondents preferred that biobanks (34%) and independent expert committees (31%) be responsible for biobank governance. And finally, Swiss most trust their own doctor, doctors in general and university researchers to keep data confidential and protected, while for-profit companies (both global and Swiss), pharmaceutical companies and health insurances are not highly trusted.

These results highlight the Swiss public's desire to be in charge of how their data and samples are used, based on their preferences for study-specific consent and personal ownership of data or samples. Switzerland strongly emphasises individual autonomy and citizen participation,[18] with direct democracy as a core feature of the Swiss political system. Therefore, these preferences are consistent with traditional cultural perceptions about individual control. However, as providing study-specific consent is often technically demanding and resource-intensive, most Swiss university hospitals have introduced broad consent ('general consent' in Switzerland).[19] Patients are asked once for consent to use their anonymised or coded data and samples for unspecified future studies.

Across the globe, consent preferences for biobank research have been well studied, with divergent results.

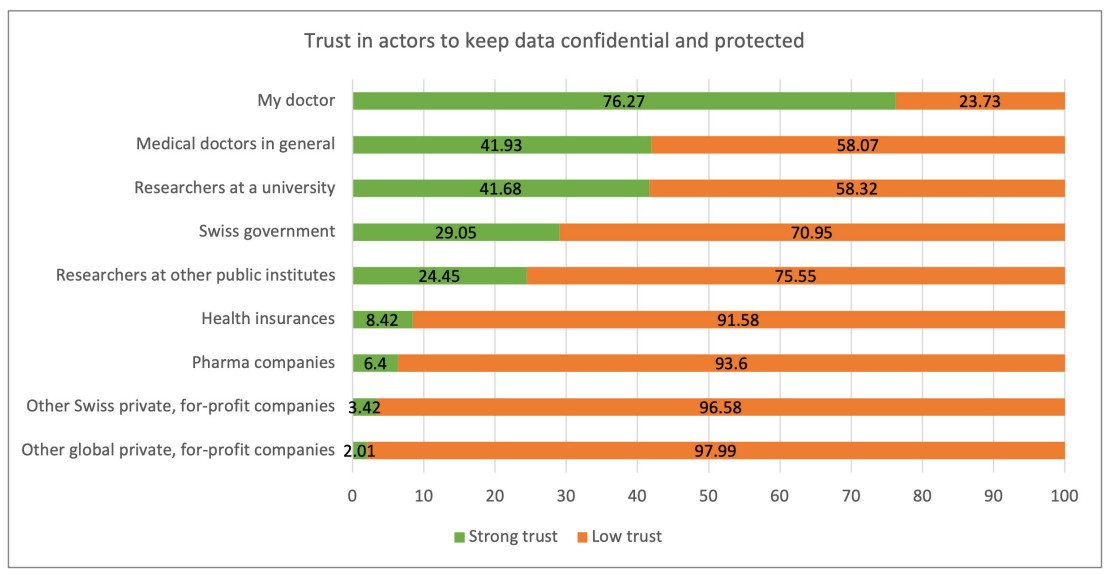

**Figure 4** Trust in actors to keep data confidential and protected. Figure presents population-weighted percentages of total respondents.

Whereas some studies of US biobanks confirm that certain types of consent increase willingness to participate in research,[20] others do not support this finding.[21–23] Most studies, however, identify a preference for broad consent,[10] broad or study-specific consent,[24] or finding both types of consent unacceptable.[4 7] In contrast, a pan-European study found that 67% prefer study-specific consent, with only 24% opting for broad consent.[25] Public preferences depend on context and culture. They are influenced by how questions are framed, which population is included (general population, research participants, patients), and the unique cultural norms and values of each country. Country-specific variation of preferences is therefore to be expected. Our finding that most Swiss respondents preferred study-specific informed consent is consistent with previously published European data.[25] It reflects the above-mentioned cultural preference to maintain a level of control when deciding for which studies data may be used.[20] It is noteworthy that existing literature about types of consent and data management is mostly US based, underscoring the importance of studying public preferences in other countries.

A further notable finding is that Swiss respondents who preferred their data be stored anonymously also frequently indicated a wish to be recontacted for each new project (34.9%). With anonymised data, the connection to recontact will be eliminated; therefore re-identifying participants is no longer possible. Such re-identification is, however, possible when data are stored in a coded form (sometimes called pseudoanonymised data). Conflicting preferences for anonymous data storage and a wish to control data use indicate a potential lack of sufficient public knowledge about data handling processes. Our survey asked about consent and storage preferences in two distinct, unlinked questions (first about consent, then about storage). Thus, we do not have insight into which choice the public would make between having their data stored anonymously and being able to reconsent to each study using their data or samples. Future research should assess preferences more specifically. However, a requirement for reconsent years after the original consent will likely result in selection bias. For example, some participants will have died and thus would be unable to reconsent.

Respondents in our study trust health professionals and institutions they know, namely their own doctors, universities and Swiss companies. They expressed the least trust in pharmaceutical companies, global for-profit companies and health insurance companies. Other studies assessing public opinions globally[26] and in the UK, USA, Australia, Canada and Singapore[2 27–32] confirm that individuals distinguish between research conducted by public and private actors. Our study identified a further distinction regarding for-profit companies: strong trust in Swiss for-profit companies increased willingness to participate in personalised health research, whereas trust in international for-profit companies had no influence. Recent scandals involving data sharing by international for-profit companies may help to account for this distrust; in 'Project Nightingale', for example, a privately held US hospital chain shared medical records with Google without notifying data subjects.[33] A similar debate occurred in summer 2021 when the British National Health Service (NHS) planned to collect, pool and share patient data held by general practitioners, as part of the so-called General Practice Data for Planning and Research (GPDPR).[34] The initiative was designed as an automatic opt-in process with a 6-week opt-out window, and no option to delete data retrospectively. Data included medical records, information on sex, ethnicity, postal code, and week and year of birth (making data easily re-identifiable), and were to be shared with healthcare and research organisations, as well as commercial entities. After its announcement, the GPDPR was widely criticised as a 'complete failure to develop a wide-ranging and far-reaching public engagement plan to communicate with the population' and was put on hold.[34] NHS Digital then initiated a consultation process and a public information campaign, to precede implementation of the initiative. This example highlights how engaging the public and ensuring the opportunity to make an informed choice about health data are crucial for sustainable and trusted health data use.

Given these insights, we might ask: how can Swiss biobanks better meet the needs and expectations of the public, and promote trust in this type of research? A first step is increasing transparency around governance mechanisms and information about research with health data and samples. We found that respondents who were unwilling to provide data or samples for research were also unsure of their preferred type of consent (80%) and were more likely to prefer their data be stored anonymously (58%). A lack of understanding of how data and samples are used for research might thus explain this disinterest in participation. It is essential that biobanks and other institutions collecting health data and samples provide accessible and understandable information via websites and brochures about how data can be reused, and the advantages and disadvantages of different types of consent for the individual and society. This would also help educate about the types of data storage, for example, anonymous versus coded data and how data type influences the ability to be recontacted or receive individual research results. Communication about future studies would promote participant knowledge about the use of their data and samples, and could help address perceived uncertainty associated with data donation and reuse, allowing participants a sense of meaningful control over their data and samples.[35] To alleviate this, Switzerland may wish to further consider developing dynamic consent models,[16] which allow participants to personalise their consent preferences. Our data suggest that dynamic consent models better mirror the preferences of the Swiss public and permit participants to align future use of their data with their personal values. Such a move towards more granular, interactive consent models allows participants

to retain greater control over their data and samples than is possible with broad consent alone.[35]

Next, strengthening trust in biobanks and research initiatives will require increased transparency about how and with whom data are shared for research. Biobanks should specify for which project data and samples will be used to any degree known, and maintain transparency about the range of future potential research that may occur, including whether data could be shared with the private sector, such as with pharmaceutical or medical technology companies.[31] In addition to listing all actors who have access to public data, the rationale and details of the cooperation should also be made transparent. In this way, potential data donors can decide for themselves whether they agree to such potential use of their data or samples before giving their consent.

As a way to increase trust, biobanks should treat the decision to donate data and samples as a process, rather than a one-time signature of the consent form.[7] Given that respondents in our study and other studies[27 30] frequently trust their own doctors most, biobanks and research initiatives should consider the possibility of doctors playing a primary role in informing patients about donating data or samples.[36] Shifting the responsibility and first point of contact to primary care would have the additional advantage of promoting broader awareness. Public trust is a dynamic construct.[37 38] It can be easily weakened if biobanks and research institutions do not meet the expectations that initially made them trustworthy in the public's perspective.[39 40] The systemic oversight approach can yield helpful mechanisms for building and maintaining public trust in increasingly complex research initiatives.[35] It requires that oversight mechanisms are adaptive to treat different data sources, and flexible to assess each intended use. Dynamic monitoring should be responsive to containing risks for data subjects.

As previously described,[3] one limitation of our study is that this survey of Swiss residents only achieved a 34% response rate. However, with the full sampling list, we could construct non-response weights by all sociodemographic characteristics (age, sex, regions and languages). Such weighting reduces the potential non-response bias, making the results reasonably generalisable to the Swiss population. In addition, this survey obtained respondents' opinions at a single point in time. As we noted elsewhere,[3] a longitudinal study could evaluate how opinions and views on this topic change over time. A further limitation of this survey is the general nature of the questionnaire, without specific examples of how the topics might apply in research; as such, respondents might not fully understand what is at stake in different scenarios when providing data or samples for research. Providing examples of different types of research in a future questionnaire could give respondents a better understanding of the dilemmas and trade-offs inherent in certain data donation contexts. For this report, we collapsed a 5-point Likert scale for levels of trust (1=no trust to 5=strong trust) to binary variables (low trust, strong trust). This

approach will have removed nuances of respondents' attitudes towards different actors. And finally, as previously noted, we asked about the type of consent and storage preferences in two separate questions, preventing us from determining which was more important to our respondents.

## CONCLUSIONS

This study reveals what the Swiss public expects from data management and governance when donating health data or biological samples for personalised health research. This study aimed to close the research gap on the Swiss public's preferences for data storage, consent, ownership and management, while also measuring trust in the ability of different healthcare actors to keep data and samples confidential and protected. These insights into public preference for data management and governance make it possible for Swiss biobanks and research institutions to consider the expectations and preferences of potential data donors. Aligning governance strategies with public expectations not only promotes trust in biobank endeavours[41]; the resulting trust also positively influences public willingness to participate in health research and biobanks.[26] Transparent communication about research, data use and implementation of consent models where participants can choose how often they would like to be asked for permission to use their data (dynamic consent) are key to increasing the willingness of individuals in Switzerland to provide their data to biobanks.

**Acknowledgements** We thank members at the Health Ethics & Policy Lab at ETH Zurich for discussions on the questionnaire development and pretesting, Christoph Freymond from the Federal Office for Statistics for providing the data sample, Stefan Wehrli and his team from the Decision Sciences Laboratory at ETH Zurich for setting up and maintaining the call centre and data collection infrastructure, Jan Kaiser for automating the digitalisation of the paper-based questionnaires and Shannon Hubbs for proofreading this manuscript. We especially thank the respondents who took part in the survey.

**Contributors** Conceptualisation—CBerlin, MZ, ME and EV. Methodology—CBrall, CBerlin, MZ, EV, ME and KEO. Software—CBerlin. Validation—CBrall, CBerlin, MZ and KEO. Formal analysis—CBrall, CBerlin, MZ, EV, ME and KEO. Investigation—CBrall, CBerlin, MZ, EV, ME and KEO. Resources—EV. Data curation—CBrall and CBerlin. Writing (original draft preparation)—CBrall. Writing (review and editing)—CBrall, CBerlin, MZ, EV, ME and KEO. Visualisation—CBrall and CBerlin. Supervision—MZ, EV and KEO. Project administration—CBrall. Funding acquisition—ME and EV. Guarantor—CBrall. All authors have read and agreed to the published version of the manuscript.

**Funding** This work was supported by Swiss National Science Foundation (SNSF) Grant 157556 to EV and through the personal research fund of EV at ETH Zurich, which covered the costs of data collection. ME was supported by a special project funding (Grant 189498) from the SNSF.

**Competing interests** None declared.

**Patient and public involvement** Patients and/or the public were involved in the design, or conduct, or reporting, or dissemination plans of this research. Refer to the Methods section for further details.

**Patient consent for publication** Not required.

**Ethics approval** We complied with data protection regulations at both research institutions (ETH Zurich and the University of Bern) and obtained ethics approval from the Ethics Committee of ETH Zurich (EK 2018-N-66).

**Provenance and peer review** Not commissioned; externally peer reviewed.

**Data availability statement** Data are available in a public, open access repository. Data are available from ETH Zürich DOI: 10.3929/ethz-b-000474690.

**ORCID iDs**
Caroline Brall http://orcid.org/0000-0002-5514-9502
Claudia Berlin http://orcid.org/0000-0003-3505-348X
Marcel Zwahlen http://orcid.org/0000-0002-6772-6346
Effy Vayena http://orcid.org/0000-0003-1303-5467
Matthias Egger http://orcid.org/0000-0001-7462-5132
Kelly E Ormond http://orcid.org/0000-0002-1033-0818

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
