## [Reviewer comments · BMJ Open]

ARTICLE DETAILS

TITLE (PROVISIONAL)	Public preferences towards data management and governance in Swiss biobanks – results from a nationwide survey
AUTHORS	Brall, Caroline; Berlin, Claudia; Zwahlen, Marcel; Vayena, Effy; Egger, Matthias; Ormond, Kelly E.

VERSION 1 – REVIEW

REVIEWER	Atutornu, Jerome Wellcome Sanger Institute, Engagement and Society, Wellcome Connecting Science
REVIEW RETURNED	14-Mar-2022

GENERAL COMMENTS	Thank you for this informative manuscript. This paper adds valuable knowledge to this field and gives a good perspective of the Swiss context. Particularly, it is clear to see in the manuscript what practical steps can be taken at the policy level to help build trust among Swiss publics. There are clear parallels with views from other jurisdictions while the unique Swiss context is appropriately highlighted.
---

REVIEWER	Blondon, Katherine University Hospitals of Geneva Department of Internal Medicine, Division of General Internal Medicine
REVIEW RETURNED	16-Mar-2022

GENERAL COMMENTS	This is the second paper about the results of this survey, focusing this time more specifically on the willingness to participant in personalized research, and preferences for data management and governance. Overall, the paper is well written, with clear introduction and results. In the methods section, I did not see any mention of how missing data is handled for these analyses. For categorical variables such as religiousness, it would have been interesting to see trend analyses. The authors mention the importance of context for individual consideration, yet the questionnaire remains very general : providing examples of different types of research could help the participant understand the stakes or difficulties of certain questions. This seems particularly important for choices about general or specific consent for studies: in theory, the participant may have an opinion, which may differ when presented with actual examples (e.g., diagnostic testing for a treatable vs non-treatable disease). Was there a reason to not choose this type of approach for the survey? The authors point out that participants wanting anonymized storage of data and wanted to be asked to consent for future
--

	studies is conflicting: yet is it really if the data is flagged for those who want to be contacted, with a second listing of contacts? Opinions change over time, and it seems important to be able to withdraw or give consent differently when these changes occur. Minor comments:  - please review verb tenses used in the methods section about public involvement. Also please specify how many members of public were involved in the initial testing (17, I believe from the other paper?) - results section, below figure 2, here are a couple suggestions : ...”to endorse the top three pieces of information...” and a bit lower, about the “most supported pieces of information” I suggest “the most relevant information” were...
--	--

VERSION 1 – AUTHOR RESPONSE

Reviewer: 1 Mr. Jerome Atutornu, Wellcome Sanger Institute, University of Suffolk	
Comments to the Author: Thank you for this informative manuscript. This paper adds valuable knowledge to this field and gives a good perspective of the Swiss context. Particularly, it is clear to see in the manuscript what practical steps can be taken at the policy level to help build trust among Swiss publics. There are clear parallels with views from other jurisdictions while the unique Swiss context is appropriately highlighted.	Thank you very much for the positive acknowledgement of our article.
Reviewer: 2 Dr. Katherine Blondon, University Hospitals of Geneva Department of Internal Medicine	
Comments to the Author: This is the second paper about the results of this survey, focusing this time more specifically on the willingness to participate in personalized research, and preferences for data management and governance. Overall, the paper is well written, with clear introduction and results.	Thank you very much for the acknowledgement of our article and the helpful feedback, which we integrated.
In the methods section, I did not see any mention of how missing data is handled for these analyses.	Thank you very much for this comment. We included questionnaires with more than 50% completeness and excluded missing data in the individual analyses, and have clarified this on page 6 in the methods section. We now also conducted a trend analysis for providing broad consent and naturally ordered

For categorical variables such as religiousness, it would have been interesting to see trend analyses. The authors mention the importance of context for individual consideration, yet the questionnaire remains very general : providing examples of different types of research could help the participant understand the stakes or difficulties of certain questions. This seems particularly important for choices about general or specific consent for studies: in theory, the participant may have an opinion, which may differ when presented with actual examples (e.g., diagnostic testing for a treatable vs non-treatable disease). Was there a reason to not choose this type of approach for the survey?	characteristics of respondents (age groups, number of household members, religiousness, educational level and health status (as in Table 1) and report the p-values for trend in the text. Thank you very much for this comment. We have added a statement to this point in the limitations section. We agree that providing examples of different types of research may have influenced the responses that participants gave, but we are unable to alter the questionnaire design at this stage. Rather, we will add this to the limitations section of the paper (page 16).
The authors point out that participants wanting anonymized storage of data and wanted to be asked to consent for future studies is conflicting: yet is it really if the data is flagged for those who want to be contacted, with a second listing of contacts? Opinions change over time, and it seems important to able to withdraw or give consent differently when these changes occur.	Thank you for this interesting thought. However, from a logistic point of view, data that is fully anonymized will no longer have the connection to recontact. The situation you describe, where one could re-identify individuals would be considered the second option of 'coded' (sometimes called pseudoanonymized) data. We have tried to make this point more clear (page 13).
Minor comments:  - please review verb tenses used in the methods section about public involvement. Also please specify how many members of public were involved in the initial testing (17, I believe from the other paper?) - results section, below figure 2, here are a couple suggestions : ...”to endorse the top three pieces of information...” and a bit lower, about the “most supported pieces of information” I suggest “the most relevant information” were... 	 - We have made these adjustments as suggested.

VERSION 2 – REVIEW

REVIEWER	Blondon, Katherine University Hospitals of Geneva Department of Internal Medicine, Division of General Internal Medicine
REVIEW RETURNED	30-Jun-2022
GENERAL COMMENTS	The paper is much improved, and I thank the authors for addressing all the points raised, and reviewing the english.